# Lower Urinary Tract Inflammation and Infection: Key Microbiological and Immunological Aspects

**DOI:** 10.3390/jcm13020315

**Published:** 2024-01-05

**Authors:** Kayle Dickson, Juan Zhou, Christian Lehmann

**Affiliations:** 1Department of Microbiology and Immunology, Dalhousie University, Halifax, NS B3H 4R2, Canada; kayle.dickson@dal.ca; 2Department of Anesthesiology, Pain Management and Perioperative Medicine, Dalhousie University, Halifax, NS B3H 4R2, Canada; juan.zhou@dal.ca; 3Department of Pharmacology, Dalhousie University, Halifax, NS B3H 4R2, Canada; 4Department of Physiology and Biophysics, Dalhousie University, Halifax, NS B3H 4R2, Canada

**Keywords:** urinary tract infection, cystitis, urethritis, urosepsis, infection, inflammation

## Abstract

The urinary system, primarily responsible for the filtration of blood and waste, is affected by several infectious and inflammatory conditions. Focusing on the lower tract, this review outlines the physiological and immune landscape of the urethra and bladder, addressing key immunological and microbiological aspects of important infectious/inflammatory conditions. The conditions addressed include urethritis, interstitial cystitis/bladder pain syndrome, urinary tract infections, and urosepsis. Key aspects of each condition are addressed, including epidemiology, pathophysiology, and clinical considerations. Finally, therapeutic options are outlined, highlighting gaps in the knowledge and novel therapeutic approaches.

## 1. Introduction

The urinary system primarily functions to filter blood and eliminate waste via urination. The urinary tract can be further subdivided into the lower (urethra and bladder) and upper (ureters and kidneys) urinary tract. The lower urinary tract is of particular interest as it is subject to several inflammatory and infectious conditions. The currently available literature lacks comprehensive reviews on the most common infections and inflammatory conditions affecting the lower urinary tract. This review will provide an overview of these conditions and associated immunological and microbiological aspects, which are critical to the development of new therapeutic options. Current standards of care, challenges to management, and emerging opportunities will also be discussed. The conditions covered include urethritis, interstitial cystitis (IC/BPS), urinary tract infections (UTIs), and urosepsis.

### 1.1. Immunological Landscape of the Urinary Tract

The lower urinary tract faces frequent exposures to pathogens, which are primarily managed through urinary voiding, which provides a mechanical defense against infection and physiological barriers. Organs in the urinary tract are characterized by the presence of a mucosal membrane. Consisting of several layers of urothelium and an overlaying glycosaminoglycan (GAG) layer, this membrane is the most impermeable in the body, providing a barrier against the external environment and urine [1]. The GAG layer plays a key role in immune responsiveness, with altered layer composition being associated with recurrent UTIs in a cohort of post-menopausal women [2]. While mechanical and physiological barriers provide good protection against pathogen exposure, additional mechanisms are required to provide a secondary layer of defense.

The urinary tract expresses a wide variety of constitutive and inducible antimicrobial peptides and proteins (AMPs). Spencer et al. identified RNase 7 as the most abundant AMPs in the urinary tract, being expressed by epithelial cells with the bladder, ureters, and kidneys [3]. Defensins and cathelicidins are also found throughout the urinary tract, which have both antimicrobial and chemotactic properties [4]. Additionally, Tamm–Horsfall protein (THP; also known as uromodulin), located solely in urine, binds FimH on uropathogens, competing with urothelial receptors to reduce bacterial adhesion to the urothelial membrane [5]. Secretory IgA serves a similar role in binding uropathogens. Despite some evidence indicating the importance of these proteins to the immune response, current evidence suggests that neither Tamm–Horsfall protein nor sIgA are able to act as biomarkers for predicting recurrent UTIs [6]. Urinary exosomes, extracellular vesicles that originate from glomerular podocytes and epithelial cells in the kidney and urinary tract, represent an emerging topic in the field as evidence suggests that they can contain a wide variety of effector molecules and may have future utility as biomarkers for conditions of the urinary tract [7]. For example, urinary exosomes collected from urine samples from healthy volunteers were shown to contain 29 antimicrobial peptides and proteins, including, but not limited to, lysozyme C, dermcidin, mucin-1, calprotectin, and myeloperoxidase [8]. These effectors provide an additional line of defense, supporting the innate immune response within the urinary tract, and may have potential for use as biomarkers of infection with additional research.

Should preliminary defenses fail, urothelial cells have additional mechanisms to support immunity against adherent uropathogens and mount a more specific response to infection. In the bladder, high levels of Toll-like receptor 4 (TLR4) are expressed on the apical surface of the urothelial cells and within vesicles containing intracellular bacteria, allowing for a rapid response to the detection of lipopolysaccharide, including the upregulation of various antimicrobial effectors, as previously described [9,10,11]. TLR5, responsible for the detection of bacterial flagellin, also has a well-established role in immunity within the bladder [12]. TLR11 is also expressed in the kidneys of mice, where it plays a role in defense against pyelonephritis, though the relevance of this receptor in humans is unknown as it is defective in humans [13]. TLR activation leads to a wide variety of immune effects, including the classical activation of nuclear factor kappa B [14]. In addition, once pathogens are detected by TLR4, the bladder urothelium possesses mechanisms to expel intracellular bacteria back into the environment and, as a last resort, will shed infected cells to prevent the spread of infection [11].

It is important to note that the immune landscape within the urethra is not as well described as immunity within the bladder. Despite similarities in tissue architecture, evidence suggests there are some tissue-specific differences, but more research is needed to fully evaluate these differences. A study by Pudney et al. demonstrated limited TLR expression throughout the male genital tract via immunohistochemistry, with the urethral epithelium expressing TLR9 [15]. When considering both resident leukocytes and the epithelium, the urethra showed the highest number of TLR+ cells overall, speaking to the importance of the urethra to immunity with the male urinary tract. It remains to be confirmed whether this distribution of TLRs is also true within the female urethra.

In addition to these unique aspects of immunity within the urinary tract, a large cohort of innate immune cells participate in the immune response. Mast cells are present in large quantities in the lamina propria of the bladder and rapidly degranulate in response to pathogens, releasing a variety of pro-inflammatory mediators [16]. Through the production of interleukin-8 (IL-8), mast cells also play a key role in the recruitment of neutrophils to the site of concern. Both tissue-resident and recruited macrophages are involved in neutrophil recruitment, with resident Ly6C- macrophages coordinating with recruited Ly6C+ macrophages to control the passage of neutrophils across the bladder basement membrane [17]. Evidence also suggests a role for natural killer and γδ T cells, though the roles for these cells are yet to be fully described [18]. Notably, the adaptive immune response is minimal when infections are confined to the bladder, but this becomes robust upon progression to the kidney [19,20].

### 1.2. Urinary Tract Microbiome

Despite the long-held belief that the urinary tract should be sterile, evidence resulting from increasingly sensitive genome sequencing techniques supports the presence of a robust urinary tract microbiome. A consensus has yet to be reached on the composition of the microbiome, but it is known to be low in biomass (<10^4^ colony-forming units per milliliter) compared to the high biomass observed in the colon microbiome (10^11^–10^12^ colony-forming units per gram) [21]. *Lactobacillus*, *Corynebacterium*, *Prevotella*, *Staphylococcus*, and *Streptococcus* are frequently identified within the microbiome of both males and females, with no statistically significant differences with respect to bacterial phyla, though proportions of specific genera may vary [22]. Additionally, there is considerable variation amongst individuals, which may inform individual responses to infections and inflammation within the urinary tract [23].

As reviewed in detail by Perez-Carrasco et al. and Chorbińska et al., current evidence suggests that the urinary microbiome or “urobiome” plays an important role in both health and disease [24,25]. Several studies have identified changes in the urinary microbiome during various conditions, including urinary incontinence, interstitial fibrosis, and tubular atrophy post kidney transplant, bladder cancer, and UTI [21,22,26,27]. In particular, changes in the abundance of *Lactobacillus* spp. have been identified in various bladder conditions [25]. Further to this, some evidence suggests that metabolites produced by the urinary microbiome play a role in immune activation in chronic kidney disease [28]. Interactions between the urinary microbiome and the immune system are yet to be fully explored, but they will likely prove important to the field of urology in the future and may play a similar role in the maintenance of homeostasis, as seen in other tissues.

## 2. Urethritis

Urethritis is broadly defined as inflammation of the urethra, which is frequently secondary to a sexually transmitted infection (STI) or a UTI. This review focuses primarily on urethritis secondary to STIs. Though urethritis can occur in either males or females, the majority of the literature focuses on afflicted males. Urethral discharge is the most common symptom of urethritis, but other non-specific symptoms can occur, including dysuria and frequency. The exact incidence of urethritis is uncertain as it is likely under-reported due to the stigma associated with STIs and the possibility of asymptomatic infection. In 2017, Canada reported over 120,000 cases of *Chlamydia trachomatis* and approximately 30,000 cases of *Neisseria gonorrhoeae*, which are both frequently associated with urethritis [29]. Risk factors for urethritis are primarily related to sexual activity, including unprotected sex and multiple partners [30].

Urethritis is classified as either gonococcal (GU) or non-gonococcal (NGU). Causative organisms and approximate prevalences are identified in Table 1. GU is primarily caused by *Neisseria gonorrhoeae*, but *N. meningitidis* has also been identified as a causative agent [31]. Various organisms can cause NGU, but *C. trachomatis* is responsible for up to half of all cases [32,33]. NGU can be etiologically complex, with diverse microbial communities often making it impossible to identify a single pathogen [34]. Co-infection is common; many patients infected with *N. gonorrhoeae* are also infected with *C. trachomatis* or less frequently with *Mycoplasma genitalium* [35]. Idiopathic infections may occur in up to 50% of all cases of NGU, which may be related to variations in the abundance of specific taxa in the urethral microbiome [36]. Despite the variety of causative agents of urethritis, symptoms, when present, are typically either discharge or dysuria [37]. The presence of leukocytes (or leukocyte esterase), indicative of inflammation in the urethra, is considered relatively sensitive for the detection of urethritis, and when combined with clinical presentation, will trigger treatment. 

Urethritis can be managed using empiric strategies, which include coverage for both GU and *C. trachomatis* [45,46]. In Canada, current recommendations indicate that a single dose each of ceftriaxone or cefixime and azithromycin should be administered to cover GU and possible co-infection with *C. trachomatis* [45,46]. Seven-day doxycycline or a single dose of azithromycin are typically used for the management of NGU, with doxycycline resulting in better efficacy against *C. trachomatis*, while azithromycin performs better against *M. genitalum* [47]. Considering the wide range of possible etiologies for urethritis, nucleic acid amplification techniques are useful for accurate diagnosis of the causative agent and prescription of appropriate antimicrobials. These techniques are recommended for persistent or recurrent urethritis [33]. Urethral smears have less utility on follow-up, as high levels of neutrophils can remain in urethral smears 2–3 weeks after the successful treatment of *C. trachomatis* infection [48]. This information suggests that the most accurate indicator of a resolved infection is a negative result from nucleic acid amplification tests, with sufficient time allowed for the clearance of the remaining non-viable pathogens.

Changes in the epidemiology of urethritis have called the use of empiric treatment strategies into question. In particular, the increasing prevalence of *M. genitalium* represents a significant challenge to the management of urethritis. Empiric treatment with a single dose of azithromycin increases the risk of macrolide resistance developing in *M. genitalium*, with resistance rates now exceeding 40% in some populations [39,49]. Estimates suggest that the overall effectiveness of azithromycin for the management of NGU may be below 80% depending on the prevalence of macrolide resistance in the area [33]. Use of a five-day regimen of azithromycin is more effective against *M. genitalium* and may be less likely to induce resistance [50]. Despite this evidence, alternative antibiotics are likely to be important moving forward. Doxycycline, typically considered a suitable alternative to macrolide therapy, is inefficient at eradicating *M. genitalium* [50]. To address these issues, clinicians should strive to identify the causative agent involved in NGU before selecting an antibiotic, particularly if *M. genitalium* is known to be prevalent in the area. Screening for mutations associated with macrolide resistance is also possible and may become increasingly important considering the threat of antibiotic resistance for the treatment of urethritis and other infections [51].

The asymptomatic nature of many STIs represents an additional challenge for the prompt diagnosis and treatment of urethritis management. In one study, urethritis was present in 16.1% of asymptomatic men screened and was associated with anal sex and multiple sex partners [52]. While the importance of these undetected infections is unclear, untreated STIs can lead to serious sequelae. In particular, *C. trachomatis* infection is associated with reproductive sequelae in both males and females [53]. Some studies have suggested that screening for asymptomatic urethritis may not be beneficial as asymptomatic patients are frequently at low risk of complications and rarely present with *C. trachomatis* infections [54,55]. The most common negative effect is transmission to a sexual partner [56]. Based on limited evidence, the Canadian Urological Association (CUA) recommends opportunistic screening for *C. trachomatis* and *N. gonorrhoeae* where feasible [57]. Screening for other less common pathogens may also be of benefit as both *M. genitalium* and Ureaplasma infections are associated with female infertility [58]. Additionally, urethritis is a risk factor for human immunodeficiency virus (HIV) infection as HIV-1 can target urethral macrophages responding to infection [59]. These additional macrophages may serve as a reservoir for disease [60]. These possible complications highlight a critical need for the early detection and management of urethritis, despite the potential for the spontaneous resolution of infection [61,62].

Innate immunity plays a key role in responding to infection, with the urethra exhibiting many of the key features of immunity at the mucosal surfaces previously described [63]. Further to this, urethritis is characterized by several unique patterns of cell-mediated responses to infection. *N. gonorrheae* infection is characterized by modest lymphocyte responses, with subsequent infections resulting in more robust lymphocyte proliferation, though this response is generally insufficient to clear the infection [64]. In vitro evidence suggests that *N. gonorrheae* infections result in enhanced T regulatory cell activity, which suppresses both Th1 and Th2 immune responses and may explain the limited protective immunity observed with repeat infections [65]. In contrast to this, a recent mathematical model examining *C. trachomatis* transmission dynamics suggests a possibility of strong, long-lasting partial immunity to reinfection. Circulating memory lymphocytes have been shown to provide robust protection against reinfection with *C. trachomatis* in a murine model of infection, and evidence in humans suggests that lower bacterial loads occur with repeat infections [66].

## 3. Interstitial Cystitis

Interstitial cystitis (IC), and related conditions such as bladder pain syndrome (BPS), are a family of inflammatory conditions affecting the bladder. There is no widely accepted nomenclature for the disorder, with the umbrella term IC/BPS encompassing conditions characterized by painful, hypersensitive bladder symptoms, including urinary frequency /urgency and nocturia [67]. IC/BPS can be divided into two distinct subtypes: Hunner type (HIC), which features characteristic lesions, and BPS, where lesions are absent (Table 2). Several other types of IC/BPS can occur, including hemorrhagic cystitis and ketamine-induced cystitis, but they are not covered in this review [68,69]. Definitions of IC/BPS used in this manuscript are reviewed in Table 2. It is important to note that these definitions are not universally accepted, which may impede research into the incidence and prevalence of these conditions.

Historically, IC/BPS was considered to be a rare condition that primarily affects females. Current evidence suggests that IC/BPS may not be as rare as previously thought as IC/BPS frequently goes undiagnosed. In 2011, the Research and Development Interstitial Cystitis Epidemiology (RICE) study estimated that 6.5% of women in the United States met a high sensitivity definition for IC/BPS, though less than 10% of these women reported an official diagnosis [70]. Further, between 1.9% and 4.2% of men screened in the RICE study met symptom criteria [71]. More research, employing definitions with both high sensitivity and high specificity, is required to fully understand the scope of these conditions.

IC/BPS is challenging from a diagnostic perspective and relies on the symptom profile (pain, urinary frequency/urgency, and nocturia) and the exclusion of other conditions (i.e., UTI) [72]. A delay in diagnosis of 5–7 years from symptom onset is common due to the many possible alternative diagnoses and knowledge deficits observed amongst primary care physicians [73,74]. Following diagnosis within the umbrella of IC/BPS, histological findings can be used to identify the disease subtype, as subtypes of IC/BPS will exhibit similar symptoms but dramatic differences in histopathology. Hunner lesions are detectable by cystoscopy, presenting as red mucosal lesions with abnormal capillary structures and scarring. This subtype, referred to as Hunner type interstitial cystitis (HIC), has a variable prevalence globally, with lesions present in 5–57% of IC/BPS patients [73]. Glomerulations may also be present in some cases, but they do not represent a unique subtype as they correlate poorly with diagnosis and can be found in asymptomatic populations [75]. Other patients who meet the criteria for IC/BPS, but do not have Hunner lesions, may be given a diagnosis of BPS [76].

Diagnosis of IC/BPS could benefit significantly from the identification of a reliable biomarker for the condition, potentially related to subtype-specific inflammatory profiles. Antiproliferative factor (APF) is linked to reduced urothelial proliferation, decreased tight junction expression, and increased paracellular permeability, which may provide a clue to the etiology of the IC/BPS [77,78,79]. APF is reasonably sensitive and specific for the detection of IC/BPS in comparison to other urogenital conditions but cannot distinguish between HIC and BPS [80]. Various cytokines may also serve as useful biomarkers. For example, increased levels of urinary CXCL10 have high specificity and modest sensitivity for detecting the presence of Hunner lesions [81]. Levels of urinary CXCL1, IL-6, and intravesical nitric oxide are also significantly increased in HIC patients in comparison to BPS patients and those either in remission or without a history of disease [82]. The macrophage migration inhibitory factor (MIF) represents another potential biomarker, which is constitutively expressed by the urothelium and released in response to inflammatory triggers, though it is also upregulated in other inflammatory conditions such as UTIs [83]. The presence of elevated HIC biomarkers may prompt further diagnostic measures such as cystoscopy to accurately identify this group of patients.

Subtypes of IC/BPS present with substantial differences in terms of inflammatory phenotypes, with particular differences in levels of mast cell infiltration. While BPS frequently presents with a moderate phenotype, HIC is characterized by chronic inflammation extending beyond the area of the Hunner lesion in 93% of HIC patients compared to 8% of patients without the presence of lesions [84]. HIC patients also presented with significant epithelial denudation, which may precede stromal inflammation. Compared to HIC, BPS patients typically exhibit a more moderate phenotype in terms of inflammation. Biopsies and gene expression profiles from these patients generally resemble a normal bladder [84,85]. Genes associated with fibrosis may be upregulated, typically in association with the presence of glomerulations [85]. These changes in gene expression may be explained by the high levels of mast cell infiltration observed in BPS compared to HIC [86]. The role of mast cells in IC/BPS is controversial and likely specific to the BPS subtype. Kim et al. suggest that in BPS, where inflammation levels are low, mast cells may promote fibrosis [86]. In the more extreme inflammatory environment of HIC, mast cells serve to activate immune cells and contribute to chronic inflammation.

The management of IC/BPS is challenging, with initial management usually focusing on conservative strategies. Patient education and dietary modifications are standard first-line therapies, with bladder training/physiotherapy and stress management when indicated [72]. Surgical resection of Hunner lesions is recommended and can produce effective symptom relief for over year, though recurrence of disease is common [87]. Hydrodistension therapy can also be employed in combination with surgical resection, resulting in improved bladder capacity and reduced voiding symptoms [88]. In one study, combined therapy provided approximately 50% of patients with long-term relief (≤3 years) [89]. BPS patients may also be treated with hydrodistension, with similar long-term efficacy despite less favorable outcomes early into treatment [90]. Beyond these key steps, precision medicine-based strategies can be employed based on symptom phenotypes (e.g., UPOINT- urinary, psychosocial, organ-specific, infection, neurological/systemic, and tenderness) [91]. This strategy looks to patient-specific symptoms to guide therapeutic decisions and emphasizes collaborative, multi-disciplinary care [92]. Current evidence suggests that phenotype-based treatment plans result in clinically significant improvements in approximately 50% of patients, though re-intervention is often still required [93,94].

Pharmacological therapies are possible in specific groups, though there are few approved therapies. A variety of medications may be employed for off-label use, including antihistamines, but this review will focus on therapeutic options specific to IC/BPS. Pentosan polysulfate (PPS) is the only approved oral treatment for IC/BPS and is thought to reduce bladder permeability by restoring the urothelial GAG layer. Oral PPS administration results in significant improvements in symptoms, as assessed by recent meta-analyses, though it may not be superior to alternative therapies [95,96]. The intravesical administration of PPS may be more effective as more of the drug reaches the target area without being subject to first pass metabolism [96]. Additionally, the co-administration of intravesical and oral PPS may be superior to oral therapy alone, though more studies are required [97]. Dimethyl sulfoxide (DMSO), which has anti-inflammatory and analgesic properties, is currently the only approved intravesical therapy for IC/BPS. DMSO is highly efficacious for some patients, but more research is needed to determine the best guidelines for use, including target patient populations and possible combination therapies [98,99].

Several novel therapies have demonstrated potential for the management of IC/BPS. Liposome-based therapies are of interest for both their own pharmacological properties and as a vehicle for other medications. The lipid bilayer of the liposome facilitates urothelial adherence, allowing the liposome to be taken into the cell via endocytosis [100]. The intravesical administration of liposomes has shown promising reductions in IC/BPS symptoms and may be as efficacious as oral PPS administration [101,102]. Further studies are needed, as liposomes have not consistently been superior to placebo treatment [103]. Cannabinoids represent another potential therapeutic strategy, directed at managing pain and inflammation associated with IC/BPS, via interactions with the endocannabinoid system. Evidence is currently limited to animal models and case reports, so further clinical studies are needed to validate these findings [104]. Cannabinoid-based therapies may be particularly useful in BPS, where pain and inflammation persist despite the lack of overt pathology in the bladder.

The progression of IC/BPS varies by patient, with some cases resolving spontaneously. A study in Asia enrolled patients who had a history of IC/BPS exceeding five years, with a mean follow-up time of over 16 years [105]. Approximately half of these patients had significant improvements in symptoms (>50%), with 12% reporting no symptoms at time of follow-up, regardless of therapeutic status. In this study, loss of follow-up was associated with the absence of Hunner lesions (BPS). This suggests that IC/BPS may decrease in severity or completely resolve over time, with the most severe cases (HIC) continuing to present for medical care. Disease severity is also linked to poor long-term outcomes, with either HIC or diffuse glomerulations experiencing only moderate improvements in symptoms despite long-term treatment [106]. In the future, studies on disease progression and resolution should attempt to stratify by subtype to determine if resolution is possible in all cases of IC/BPS or is subtype-specific. It is also important to clarify if these improvements are true disease resolution or the result of lifestyle modifications and improved self-management over time.

A variety of co-morbidities can occur with IC/BPS, suggesting that IC/BPS may be part of a more diffuse systemic disorder. Approximately two thirds of women in the RICE cohort reported at least one non-bladder condition [107]. Conditions considered to be functional somatic syndromes, such as fibromyalgia, irritable bowel syndrome (IBS), chronic pain and fatigue, and migraines, are frequently co-morbid with BPS, with 90% of patients exhibiting characteristics of functional somatic syndrome [108,109]. These conditions are also risk factors for IC/BPS development as female patients suffering from depression, chronic pain conditions, malaise, and inflammatory disorders are more likely to develop IC/BPS [110]. Dysregulated endogenous pain control is a feature in functional somatic syndromes and has also been observed in IC/BPS patients, who exhibit reduced pain tolerance compared to healthy individuals [111]. These patients also exhibit blunted cortisol responses and excessive TLR4-mediated inflammatory responses, though a temporal relationship has yet to be established [112]. A systemic disorder affecting multiple organs might explain the painful symptoms associated with BPS that occur in the absence of overt end organ pathology. This supports theories that suggest that HIC and BPS should be considered as two etiologically distinct disorders, but additional research is necessary to fully understand the relationship between somatic changes and IC/BPS.

Determining the underlying etiology of IC/BPS remains the biggest challenge associated with the disorder, with current knowledge suggesting that HIC and BPS likely have different etiologies. Pathophysiological changes resulting in altered bladder permeability represent a compelling theory for the origins of HIC. A recent multi-site study determined that patients with IC/BPS had significantly elevated levels of cationic toxins within their urine, though the study did not stratify by disease subtype [113]. Increased levels of cationic toxins can damage the bladder and increase permeability. This theory is supported by bladder damage and inflammation associated with the abuse of the cationic molecule which occurs with ketamine abuse [114]. Alterations to the GAG layer or neuromodulator expression within the bladder also alter permeability and may represent possible etiologies [115]. Nitric oxide, a key neurotransmission mediator within the bladder, is known to be upregulated in HIC [116]. With respect to BPS, it seems likely that disease is related to neural dysregulation due to the connections to functional somatic syndromes and additional inflammatory dysregulation observed in BPS patients.

As relatively little is known about the etiology of IC/BPS, selecting an appropriate animal model is challenging. Models may be bladder-centric, where a noxious substance is applied to the bladder, or they may be more complex, relying on CNS alterations or stress induction [117]. Bladder-centric models are common, but they frequently result in pathology that is not seen in IC/BPS. For example, intraperitoneal cyclophosphamide results in IC/BPS symptoms within 24 h, but additional features of this model make it more suited to modeling hemorrhagic cystitis. Other non-selective agents, such as LPS, have limited utility as they cause damage extending beyond the urothelium. Additionally, many of the models used fail to capture the chronic nature of IC/BPS, where symptoms may vary in severity over time, or any associated systemic conditions. Where the etiology of IC/BPS remains poorly understood, it remains necessary to conduct research in various animal models which recreate different aspects of the disease. Clinical research, with stratification by disease subtype, also remains critically important for elucidating pathophysiology and mechanisms of disease.

Recently, there has been a focus on potential links between UTIs and the development of IC/BPS. By definition, diagnosis of IC/BPS requires the exclusion of UTI, but infection may serve as an initiating event in some cases. A recent study suggested that a causative pathogen may be revealed with the use of advanced culture techniques (e.g., specialized media and long incubation times) as some patients report relief following antibiotic administration [118]. Evidence is mixed, however, as the limited differences in urinary microbiome observed when comparing to healthy cohorts suggests that there may not be a major microbial contributor [119]. Recurrent UTIs may also impact the likelihood of IC/BPS development. In one study, IC/BPS patients reported experiencing significantly more problematic urinary symptoms during childhood than healthy individuals, including recurrent UTIs and urinary urgency [120]. Recurrent UTIs and IC/BPS are also both associated with bladder hypersensitivity [121]. These links suggest that there may be an underlying defect that contributes to both recurrent UTIs and the development of IC/BPS. For example, patients with recurrent UTIs have defects in the expression of proteins involved in urothelial cell proliferation, cytoskeleton, and barrier function, which could also contribute to IC/BPS [122]. This is further supported by evidence which suggests that both recurrent UTIs and IC/BPS can benefit from similar treatments, with both conditions responding favorably to antioxidative and anti-inflammatory therapies (e.g., cranberry and pine bark extracts) [123]. More longitudinal studies are needed to fully elucidate this relationship.

## 4. Urinary Tract Infections

Urinary tract infections (UTIs) can occur within any portion of the urinary tract and are commonly classified as infections of either the upper or lower urinary tract. The most important sites of infection are the bladder and the kidneys, referred to as bacterial cystitis and pyelonephritis, respectively. Definitions and associated symptoms of UTIs are provided in Table 3. UTIs are generally classified as uncomplicated, which occur in a healthy, premenopausal female, or complicated, which involves the presence of additional risk factors (advanced age, pregnancy, male sex, etc.). Uncomplicated cystitis occurs at a rate of 0.5 infections per person annually in young, sexually active females, with approximately half of all females experiencing at least one UTI in their lifetime [124,125]. The incidence of pyelonephritis is significantly lower; approximately 28/10,000 females between the ages of 18 and 49 experience pyelonephritis annually, with 7% of these cases requiring hospitalization [125]. Aside from the high incidence seen in young adult females, the incidence of UTIs increases gradually with age for both males and females, though the misclassification of asymptomatic bacteriuria (ASB; presence of bacteria without clinical symptoms) likely inflates these values. Differences in incidence between the two sexes also narrow with age. In a cohort over the age of 85, UTI incidence was determined to be 0.13 per person–year for females and 0.08 per person–year for males [126].

Premenopausal, sexually active females have the highest risk of contracting a UTI, but various other factors may enhance susceptibility. The combination of risk factors at play determines whether the UTI is classified as uncomplicated or complicated. Additional risk factors for uncomplicated UTIs include the use of certain contraception methods (e.g., spermicide use), vaginal infection, diabetes, and obesity [127]. A family history of UTIs is strongly associated with recurrent UTIs and pyelonephritis, suggesting a strong genetic component to susceptibility [128]. This may be related to enhanced susceptibility to uropathogen colonization, such as uropathogenic *E. coli* (UPEC), via the upregulation of UPEC receptors in the urinary tract, though limited evidence is available to support this theory [129]. Complicated cystitis has additional specific risk factors, which are primarily related to general immune status and anatomical/functional abnormalities affecting the urinary tract (e.g., presence of vesicoureteral reflux). Though primary immunodeficiencies predispose an individual to infections, UTIs make up only a small proportion of infections observed in these individuals [130]. This likely reflects the effectiveness of the many redundant defense systems observed within the urinary tract.

**Table 3 jcm-13-00315-t003:** Definitions and diagnostic factors associated with upper and lower tract UTIs.

Classification	Definition	Symptoms	Diagnosis
Bacterial Cystitis	Lower UTI affecting the bladder	Dysuria, urgency, frequency, suprapubic tenderness	>10^5^ CFU/mL in urine culture [131] <10^3^ CFU/mL typically considered contamination, unless collected by suprapubic aspiration or there are other clinically relevant findings [132]
Pyelonephritis	Upper UTI infecting the kidneys	Systemic symptoms; fever, chills, flank pain	+/− positive urinalysis with symptoms of systemic infection
Uncomplicated UTI	Occurs in non-pregnant females with no complicating anatomical abnormalities	As described for uncomplicated/complicated UTI	Positive urine dipstick or >10^5^ CFU/mL in urine culture (midstream specimen)
Complicated UTI	Factors associated include maleness, advanced age or childhood, anatomical/functional abnormalities, pregnancy, catheter use, immune compromised state, or atypical pathogens	As described for uncomplicated/complicated UTI	>10^5^ CFU/mL in urine culture (midstream specimen) 10^3^–10^4^ CFU/mL in catheterized samples [132]
Asymptomatic bacteriuria	Presence of bacteria in urine in the absence of symptoms [133]	No symptoms	CFU threshold met in urine but no symptoms [131]
Recurrent UTI	3+ UTIs within 12 months, or 2+ UTIs within six months [134]	As described for uncomplicated/complicated UTI	As described for uncomplicated/complicated UTI

Note. The number of CFU/mL considered sufficient for the diagnosis of UTI is dependent on the method of urine collection. A diagnosis of UTI may be supplied without meeting these criteria, if other clinical factors suggest a UTI, at the discretion of the clinician.

Catheterization is another major risk factor for UTIs, with these infections referred to as catheter-associated UTIs (CA-UTIs). CA-UTIs are the most common healthcare-associated infection in the United States and are defined as the occurrence of UTI symptoms without another source of infection and more than 10^3^ CFU/mL in urine from a patient who is catheterized or whose catheter was removed recently (<48 h) [135,136]. Importantly, asymptomatic bacteriuria can occur with a prevalence of up to 100% in catheterized patients (CA-ASB), but treatment is not recommended for this [131,137]. Some evidence suggests that intermittent and indwelling catheterization have similar rates of CA-UTI, but intermittent catheterization is strongly recommended for long-term use by the Infectious Diseases Society of America because of the quality of life benefits [138,139]. The pathogens responsible for CA-UTI are largely similar to those of complicated UTI, as described in Table 4, but pathogens with the ability to form biofilms, such as UPEC, are of particular importance. One study determined that 73.4% of pathogens isolated from patients with CA-UTI were biofilm producers [140]. High rates of antibiotic resistance were observed amongst these strains, making treatment of these infections complex. Because of these high rates of resistance, good antimicrobial stewardship is particularly critical for the management of these infections and the preservation of current antibiotic options.

UTIs result from the exposure of the urinary tract to uropathogens present in the intestinal flora, typically via fecal matter contaminating the periurethral area. UPEC, a facultative pathogen when outside of the gut, causes 75% off all uncomplicated UTIs [141]. Other uropathogens, and their prevalence in complicated and uncomplicated cystitis, are outlined in Table 4. Pathogens seen in pyelonephritis resemble those seen in bacterial cystitis, as most kidney infections ascend from the bladder, though non-UPEC pathogens are more prevalent. Hematogenous spread to the kidneys is rare, but it has been associated with Group B Streptococcus infection [142]. Typical pathogenesis for an ascending UTI is illustrated in Figure 1. Though rare, systemic spread can occur secondary to a UTI, resulting in urosepsis. This is discussed in detail in the following section.

**Table 4 jcm-13-00315-t004:** Prevalence of uropathogens.

Pathogen	Prevalence		Notes
	Uncomplicated Cystitis	Complicated Cystitis	
UPEC	75	65	
*Klebsiella pneumoniae*	6	8	
*Staphylococcus saphrophyticus*	6	-	Young women are selectively susceptible to colonization [143]
*Enterococcus* spp.	5	11	Low-grade symptoms; role in polymicrobial UTI [144]
Group B Streptococcus	3	2	Associated with diabetes, chronic renal failure, and post-partum infection [145]
*Proteus mirabilis*	2	2	Associated with formation of bladder/kidney stones, diabetes [146]
*Pseudomonas aeruginosa*	1	2	Opportunistic pathogen in patients susceptible to UTI [146]
*S. aureus*	1	3	Associated with catheterization; frequently methicillin-resistant [147]
*Candida* spp.	1	7	Nosocomial opportunistic pathogens [148]

Note. Prevalence data sourced from Flores-Mireles et al. [141].

UTIs are commonly diagnosed in the clinic via urinary dipstick, which relies on the presence of leukocyte esterase and nitrites within urine, and is a relatively sensitive and specific test [149]. The diagnosis of pyelonephritis is made based on a positive urinalysis with additional clinical findings indicative of a systemic infection. Treatment is generally empiric, based on epidemiological data regarding pathogen prevalence. If complications are suspected, or systemic symptoms are present that may indicate pyelonephritis, urine cultures are performed to guide treatment decisions. Positive cultures cannot be relied on as the sole metric for pyelonephritis, however, as a significant portion of patients with pyelonephritis and/or associated complications visible via imaging techniques will have negative cultures [150]. This could be related to prior antibiotic use, atypical pathogens, or a low bacterial burden in the sample.

Care should be taken to distinguish between asymptomatic bacteriuria (ASB) and UTI during clinical assessment. ASB is defined as greater than 10^5^ CFU/mL within the urine in the absence of symptoms of infection. ASB is a relatively common condition, occurring in 2.7% of young females between the ages of 15 and 24. The incidence of ASB is dramatically increased in the elderly, particularly those living in long-term care, where it has a prevalence of up to 20% in males and 50% in females [133,151]. This heightened prevalence is related to the increased presence of co-morbidities in the elderly population (e.g., diabetes, intermittent catheterization, etc.). The Infectious Diseases Society of America recommends against screening for and treating ASB, except in the case of pregnancy or impending urological procedures [133]. The overtreatment of ASB represents a major challenge in the field currently as it can contribute to increased rates of antimicrobial resistance. A positive urine culture may trigger the prescription of antibiotics by clinicians, despite no other signs of infection. A set of multifaceted guidelines for the ordering of urine cultures and treatment of ASB were able to significantly improve management practices, with the most benefit occurring in long-term care wards [152]. The widespread implementation of such guidelines should be considered as part of good antimicrobial stewardship.

In the case of uncomplicated cystitis, empiric treatment strategies can be employed. Nitrofurantoin, trimethoprim sulfmethoxazole (TMP-SMX), and fosfomycin are considered to be first-line therapies [153]. For complicated UTIs, first-line therapies include cefixime, amoxicillin–clavulanate, TMP-SMX, and fluoroquinolones. Antibiotic therapy decisions should be guided by information on local resistance patterns. TMP-SMX, previously considered to be the drug of choice for UTIs, now has resistance rates exceeding 20% in many areas [154]. Symptoms should begin to improve within 48–72 h, unless antibiotic coverage is inappropriate. One study identified that up to 68% of outpatients were prescribed inappropriate treatment regimens, resulting in high rates of UTI recurrence and reinfection [155]. Catheter use and advanced age were risk factors for inappropriate treatment.

Though antibiotic administration is the current standard of care, several studies suggest that the majority of UTIs can resolve on their own without progressing to pyelonephritis or urosepsis. Various trials have compared antibiotic therapies to either placebo or symptomatic treatment. In one study, though nitrofurantoin was significantly more effective at eradicating uncomplicated UTI, over half of the placebo-treated patients experienced symptom improvement and cure after seven days [156]. In a similar study, patients with uncomplicated cystitis treated with ibuprofen only did not experience significantly different rates of recurrence or pyelonephritis up to six months after initial treatment compared to those managed with standard antibiotics [157]. Other non-antibiotic therapies, including cranberry products, estrogen, probiotics, vitamins, and immunotherapy, have demonstrated some utility, but they are still inferior to antibiotics. A recent systematic review indicated that the resolution of symptoms occurs within 7 days in 26–75% of patients who received delayed or placebo-based therapy and 70–83% of patients who received non-steroidal anti-inflammatory therapy [158]. As of yet, the herbal phytodrug Canephron N is the only non-antibiotic therapy to prove non-inferior to antibiotics in clinical trials [159]. Canephron N therapy potentially resulted in the increased development of pyelonephritis (five cases vs. one in the antibiotic comparator group), but more well-designed clinical trials are required to validate this finding. This evidence suggests that non-antibiotic management strategies could be employed prophylactically to reduce recurrent UTIs. These non-antibiotic management strategies may target some of the specific adaptations observed in various uropathogens.

Uropathogens possess several unique adaptations to facilitate pathogenesis. The ability to ascend and adhere within the urinary tract is critical for the establishment of infection. Flagella facilitate ascension through the urethra and ureters, and though not essential for virulence, the presence of flagella confers a significant fitness advantage to UPEC in terms of persistence within urine, and they are important for biofilm formation [160]. Flagella are also important to other uropathogens, e.g., flagellar mutations in *P. mirabilis* lead to a reduction in virulence of at least 100-fold [161]. Adherence and colonization are primarily facilitated by fimbrial adhesins. Type 1 fimbriae, specifically the FimH adhesin, are associated with bladder colonization, but they synergize with P-fimbriae to promote kidney colonization [162]. Type 1 fimbriae are also present in *K. pneumoniae*, but FimH has a different binding specificity compared to UPEC, which results in reduced adhesion and lower urine titers [163]. The PapG adhesin on the tip of the fimbriae is not essential to renal colonization despite its widespread presence in pyelonephritis pathogens [164,165]. This is likely attributable to the redundancy observed within UPEC fimbrial systems, where several types of fimbriae can bind different sites within the kidney [166]. Other uropathogens possess different adaptations to facilitate adherence in the urinary tract, such as type IV pili in *P. aeruginosa* and a variety of adhesins in *S. aureus* [167,168]. Strong adherence contributes to the formation of a biofilm within the bladder, and the spread of infection.

UPEC adhesion also plays a role in the development of recurrent infections. Recurrent UTI is defined by the occurrence of either two infections in a six-month period or three infections over a year and occurs in upwards of 20% of patients [169]. The formation of intracellular bacterial communities (IBCs) facilitates these persistent infections. These communities are undetectable via standard urine culture, though they may be associated with low levels of CFU/mL that do not meet the threshold for UTI diagnosis as bacteria are released back into the bladder lumen from mature IBCs [170]. IBCs have been detected by microscopy within exfoliated urothelial cells in the urine [171]. Though UPEC are primarily associated with IBCs, *E. faecalis* and *K. pneumoniae* can also form IBCS experimentally [172,173]. IBCs form as a result of the internalization of fusiform vesicles triggered by the attachment of UPEC to urothelial cells [11]. Urothelial cells possess a specialized mechanism to detect these invading pathogens, mediated via Toll-like receptor 4, which triggers exocytosis to expel pathogens. Some UPEC escape the vesicle into the cytoplasm where they can multiply and form IBCs. Mature IBCs have a biofilm-like structure, where they can persist long-term as they are protected from both host immune responses and antibiotics. UPEC can also establish quiescent intracellular reservoirs (QIRs) in the deeper transitional cells of the bladder, which can lie dormant for months until an event such as epithelial turnover results in reactivation [174].

One of the most pressing issues in the landscape of UTI management is the increasing rate of antimicrobial resistance. A recent meta-analysis identified that the pooled prevalence of resistant strains of UPEC exceeded 50% in children with UTIs globally [175]. Resistance rates were particularly high in countries where antibiotics are likely to be available over the counter. This trend is echoed in other age groups. High levels of resistance to both ampicillin and nitrofurantoin were observed in the elderly in Norway, regardless of whether they lived in nursing homes or in the community [176]. The rise in antibiotic-resistant strains of UPEC has the potential to limit treatment options for UTIs in the future. While additional research into novel antibiotics and adjunct therapies may be helpful, a lack of new bacterial targets and waning interest from pharmaceutical companies makes this avenue significantly less promising. Good antimicrobial stewardship likely represents the best way forward, to slow antibiotic resistance and preserve current therapeutic options.

## 5. Urosepsis

Sepsis is defined as life-threatening organ dysfunction caused by a dysregulated host response to infection, with septic shock being further defined as sepsis with the presence of severe circulatory and cellular/metabolic abnormalities [177]. The pathogenesis of sepsis is highly complex and includes dysregulated inflammatory and immune responses, microcirculatory changes, and coagulopathy [178]. In 2014, approximately 1.7 million adults were hospitalized for sepsis in the United States, with 15% of these patients dying in hospital and a further 6.2% being discharged to hospice [179]. Since 1990, significant decreases have occurred in both the incidence and mortality of sepsis [180]. Despite this, sepsis still represents a major healthcare challenge globally.

Urosepsis occurs when the source of the initiating infection is within the urinary tract. Though UTIs rarely progress to sepsis, in the United States, approximately 30% of sepsis cases have urological origins [181]. Urosepsis may be community-acquired or nosocomial in nature. Both anatomical abnormalities and conditions affecting urodynamics are considered risk factors for progression to pyelonephritis and ultimately to urosepsis [182,183]. Urosepsis can also occur secondary to urological procedures and becomes significantly more likely in patients which immune-compromising co-morbidities such as diabetes mellitus [184]. The role of catheters in the development of urosepsis remains unclear, with studies reporting mixed results with respect to catheter usage as a risk factor for urosepsis [185]. With community-onset urosepsis, risk factors include urinary tract disorders and the prescription of inappropriate antibiotics [186]. The papGII gene, for the adhesin subunit of the P-pilus in UPEC, is also a significant risk factor for progression from UTI to urosepsis [187].

Though urosepsis is the most common type of sepsis, it frequently has a significantly better prognosis than sepsis of other origins. Rhee et al. determined that sepsis had an overall mortality rate of approximately 21% in the United States in 2014 [179]. Despite the early appearance of organ dysfunction, urosepsis patients enrolled in a small observational study had a significantly better 28-day mortality rate when compared to other septic patients (6% vs. 37%) [188]. It should be noted that the demographic factors of urosepsis patients are often different from patients with other types of sepsis. In the study by Qiang et al., the urosepsis patients were significantly younger than the control group and had less co-morbidities [188]. Urosepsis develops secondary to a surgical procedure. In an older population, mortality due to urosepsis is associated with functional dependency, number of co-morbidities, and low serum albumin [185]. The reduced mortality observed with urosepsis is likely related to favorable patient characteristics such as younger age, though ease of source control is probably a contributing factor.

Sepsis is a complex syndrome, which includes a variety of signs and symptoms that can complicate prompt identification. In a clinical setting, sepsis is typically identified in patients with a suspected or confirmed infection by screening for signs of organ dysfunction using the quick sequential organ failure assessment (qSOFA) at the bedside, followed by the complete SOFA, which incorporates more parameters [177]. With respect to urosepsis, it can be challenging to differentiate between febrile UTI (pyelonephritis) and urosepsis, as urine/blood cultures are frequently negative, particularly if antimicrobial therapy has been initiated for an underlying UTI or as prophylaxis for surgical procedures [189]. A low procalcitonin/albumin ratio has been proposed as a relatively sensitive and specific early indicator of urosepsis, which may allow for the accurate distinction of febrile UTIs and urosepsis and identify patients who are be prone to septic shock [190]. Serum C-reactive protein has also been identified as a risk factor for shock development in urosepsis [191]. While some progress has been made, additional work is needed to develop reliable biomarkers for urosepsis, particularly considering the importance of early treatment.

The rapid diagnosis and administration of antibiotics are critical to outcome, with even brief delays in medical contact and antibiotic administration being associated with increases in poor outcomes [192,193]. Though compelling, these data must be interpreted cautiously as the exact time of the onset of sepsis is often unknown. These data do appear to support the rapid administration of empiric antibiotics, with changes made to antibiotic coverage as dictated by culture results, if available. In addition to this, a retrospective study determined that the administration of antibiotics within the first hour of documented hypotension, which indicates the onset of shock, was associated with increased odds of survival to discharge [194]. This may be valuable knowledge with respect to the management of critically ill patients, whose cardiovascular parameters are being constantly monitored, but this is likely less useful for patients presenting from the community with possible urosepsis.

Standard of care for septic patients, as outlined in the 2016 Surviving Sepsis Guidelines, includes control of infection at the source, early administration of appropriate intravenous antibiotics, fluid resuscitation with crystalloids/hydroxyethyl starches, hemodynamic support (i.e., vasopressors), and ventilation as necessary [195]. Other therapies can be employed based on the specific patient, such as renal replacement therapy or stress ulcer prophylaxis. Source control is particularly important for urosepsis as abnormalities of the urinary tract represent a complicated factor in cases of urosepsis. Up to 10% of patients present with an obstruction, which is associated with increased mortality [196]. Imaging is recommended to aid in the identification of possible abnormalities for correction [197].

Pathogens involved in urosepsis typically resemble those involved in the development of UTIs. A recent study identified *E. coli*, *Proteus* spp., *Enterococcus* spp., and Gram-positive cocci as primary causative agents, with *P. aeruginosa* and *C. albicans* occurring in patients who were significantly immunocompromised [188]. As in UTIs, *E. coli* represents the most common pathogen in urosepsis. This knowledge can be used to guide empiric therapy in the absence of culture results. Appropriate antibiotics for urosepsis include intravenous ceftriaxone or gentamicin, with the possible addition of ampicillin [153]. Recent advances in laboratory technology have made it possible to identify uropathogens in three hours, via mass spectrometry, in comparison to the 24 h processing time required by culture-based techniques [198]. This reduced turn-around time will allow for the validation of appropriate antibiotic therapy, or adjustment of coverage as necessary. This is important to both the patient and the healthcare system, as initial inappropriate antibiotic therapy is associated with both increased readmission to the hospital and increased cost of care [199]. *P. aeruginosa* infection in particular is associated with inappropriate therapy, as ceftriaxone has limited activity against *Pseudomonas* spp. [200]. This pathogen is associated with increased length of hospital stay compared to other uropathogens [200].

Despite the relatively good prognosis for urosepsis overall, the causative pathogen can have a significant impact on outcomes. The emergence of multi-drug resistant (MDR) pathogens represents a major challenge for the treatment of urosepsis. Extended-spectrum β-lactamase (ESBL) producing Enterobacteriaceae are MDR pathogens with resistance to various β-lactam drugs and third-generation cephalosporins. As of 2016, all ESBL-producing E. coli collected for the CANWARD study were susceptible to colistin, and most were susceptible to carbapenems (98–99.8%) and nitrofurantoin (92.5%) [201]. Recently, a Canadian hospital noted an incidence of 19.4% for ESBL urosepsis, with chronic renal insufficiency, recent travel to an endemic region, a first language of Punjabi or Hindi (representing possible contact with endemic regions), or male sex as important risk factors. These patients had a significantly worse prognosis in terms of mortality compared to urosepsis caused by other pathogens [202]. This is likely attributable to the delayed initiation of appropriate antimicrobial therapy [203]. Though Canada still has many effective antibiotics for the treatment of ESBL urosepsis, good antimicrobial stewardship and the prevention of urosepsis will likely be critical to preventing the development of additional MDR pathogens. Carbapenem-resistant Enterobacteriaceae (CRE) are an emerging threat globally, with 9.1% of K. pneumoniae infections being carbapenem-resistant in New York, which is currently considered to be the global epicenter of CREs [204]. CRE infection is associated with significantly higher rates of both septic shock and mortality [205]. Limited information is available on the prevalence of CREs in urosepsis, but the nature of this pathogen suggests that this may become a concern in the future.

The immune dysregulation that occurs during sepsis is a major contributor to mortality [206]. Initially, sepsis is characterized by hyperinflammation. This pro-inflammatory phase is followed by immunosuppression, though it can be challenging to discern where patients lie along this spectrum. Up to a third of patients in the late phase of sepsis (>9 days) will develop a secondary infection related to immunosuppression [207]. Additionally, patients who die of sepsis are more likely to exhibit markers of immunosuppression, as assessed by immunohistochemistry, cytokine secretion assays, and immunophenotyping than those who die of non-sepsis causes [208]. Urosepsis appears to cause early organ dysfunction, suggesting that the progression of sepsis is rapid in these patients [188]. When taken in combination with low mortality rates, this may suggest fewer patients progress to an immunosuppressive phase in urosepsis. Few trials address urosepsis specifically, so further research will be required to determine the specifics of disease progression in urosepsis. It seems plausible that anti-inflammatory therapies may be more successful in this group of patients than they have been for sepsis overall, particularly if administered at the earliest possible timepoint.

## 6. Conclusions

The urological conditions discussed in this essay include urethritis, IC/BPS, UTIs, and urosepsis. These conditions are vastly different in terms of etiology and standard of care, but they have some shared challenges. Accurate diagnosis, including distinguishing between these conditions and the identification of causative pathogens (where applicable), remain difficult. Diagnostic challenges delay appropriate therapeutic interventions, to the detriment of patients, and may contribute to the development of antibiotic resistance. Additional research, both at the bench and in the clinic, is necessary to better our understanding of the etiology and pathology of some conditions. There is a wealth of knowledge available which may have implications for the management of these conditions in the future, but knowledge translation remains a major barrier to implementation. Challenges related to antibiotic resistance can be managed through increased education, the implementation of prophylactic measures, and good antimicrobial stewardship.

## Figures and Tables

**Figure 1 jcm-13-00315-f001:**
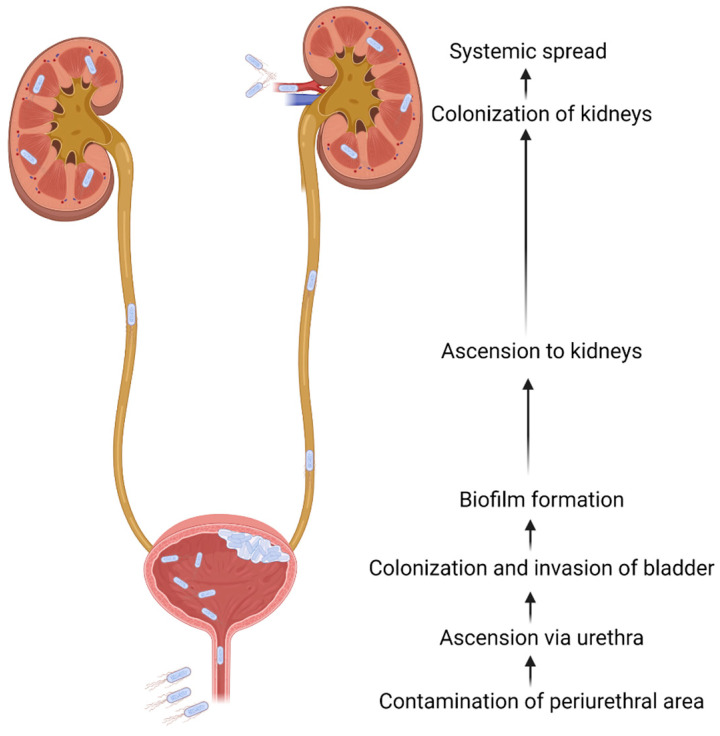
Pathogenesis of UTIs. Most UTIs result from the contamination of the periurethral area with pathogens from the intestinal flora. These pathogens ascend the urethra and then colonize the bladder. This leads to biofilm formation and intracellular invasion. The infection progresses to pyelonephritis if the pathogens ascend the ureters and colonize the kidneys. Though rare, systemic spread can occur in the kidneys, leading to urosepsis.

**Table 1 jcm-13-00315-t001:** Pathogens involved in urethritis secondary to STIs in order of prevalence.

Classification	Pathogen	Prevalence	Notes	References
GU	*Neisseria gonorrhoeae*	30%	-	[32]
	*Neisseria meningitidis*	4%	May account for up to 20% of GU cases	[31,32]
NGU	*Chlamydia trachomatis*	11–50%	-	[32,33]
	Mycoplasmas	5–22%	Primarily *M. genitalium*, but *M. hominis* possible	[32,35,38,39]
	Ureaplasmas	9–20%	NGU primarily associated with *U. urealyticum*	[32,40]
	*Trichomonas vaginalis*	1–3%	-	[32,41,42]
	Adenoviruses	2–16%	Associated with concurrent conjunctivitis	[32,43]
	Herpes simplex virus	2–7%		[32]
	Idiopathic	≤50%	Absence of common pathogens; may include *Haemophilus influenzae* or *Mycoplasma penetrans* in males	[44]

**Table 2 jcm-13-00315-t002:** Definitions of IC/BPS and subtypes.

Term	Description
IC/BPS	Umbrella term: condition characterized by painful, hypersensitive bladder symptoms including urinary frequency/urgency and nocturia with the absence of other explanatory conditions (e.g., UTI)
HIC	IC/BPS with distinct Hunner lesions as visible by cystometry
BPS	IC/BPS without Hunner lesions; glomerulations may be present

Note. Definitions sourced from Homma et al. [67].

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
