# Peer review of "Lower Urinary Tract Inflammation and Infection: Key Microbiological and Immunological Aspects"

_jcm, 2024, doi:10.3390/jcm13020315_

Round 1

Reviewer 1 Report

Comments and Suggestions for Authors

This is a nice narrative review of a broad topic of different conditions of the urinary system. 

- Why is the title treatment of? The review is about all key aspects. 

- Definition of UTI are confusing and without reference.

The presence of risk factors for development of a complicated UTI is not the same as having a complicated UTI (pyelo or urosepsis). 

There is evidence that urine culture < 10^3 could still be a cystitis.

Table 3 Diagnosis part is probably for both cystitis and pyelo, but this is not clear. 

Why is the definition of cystitis other than uncomplicated UTI? This is basically the same. 

- For CA-UTI, why are the authors suggesting CIC instead of long-term use? There is also evidence that UTI frequencies does not differ.  

I miss CA-ASB in the part of CA-UTI 

- For the manuscript table 4 should be about CA-UTI. However, what is uncomplicated cystitis in patients with a catheter? Could the author clarify this part, and also include CA-UTI in the title of table 4 if this is about uropathogens in CA-UTI.

- In the pathogenesis part I miss a part about lactobacilli and the urinary microbiome.

- Where is figure 1?

- Page 5 referral to Table 1 is not correct.

- For clearance, it would be optional to included subtitles, such as treatment / pathogenesis

Comments on the Quality of English Language

Overall, good quality of English language. 

Author Response

Thank you for providing these helpful comments.

- Why is the title treatment of? The review is about all key aspects. 

Response: We agree with this comment and have revised the title to be more comprehensive. The title now reads as follows: Lower Urinary Tract Inflammation and Infection: Key Microbiological and Immunological Aspects

- Definition of UTI are confusing and without reference.

Response: We have revised the language slightly in this portion of the paper, and have added a reference.

Werneburg, G.T.; Rhoads, D.D. Diagnostic Stewardship for Urinary Tract Infection: A Snapshot of the Expert Guidance. Cleve Clin J Med 2022, 89, 581–587, doi:10.3949/ccjm.89a.22008.

The presence of risk factors for development of a complicated UTI is not the same as having a complicated UTI (pyelo or urosepsis). 

Response: We agree with this, and have revised the paragraph on risk factors to more clearly indicate that the risk factors described are primarily for uncomplicated UTI.

There is evidence that urine culture < 10^3 could still be a cystitis.

Response: We have revised Table 3 to include some additional details and a reference to clarify this point.

Tullus, K. Low Urinary Bacterial Counts: Do They Count? Pediatric Nephrology 2016, 31, 171–174, doi:10.1007/s00467-015-3227-y.

Table 3 Diagnosis part is probably for both cystitis and pyelo, but this is not clear. 

Response: We have updated the table caption to clearly state that the table encompasses both lower and upper tract UTIs. We have also revised the accompanying paragraph to clearly reflect this.

Why is the definition of cystitis other than uncomplicated UTI? This is basically the same. 

Response: Cystitis refers only to inflammation, so we have revised the terminology in the section to clearly state bacterial cystitis, which implies the presence of an infection as the cause of the inflammation.

- For CA-UTI, why are the authors suggesting CIC instead of long-term use? There is also evidence that UTI frequencies does not differ.  

Response: We thank the reviewer for this insight. We have based our review on available clinical guidelines, which typically recommend intermittent catheterization when possible. This is partially due to benefits that the patients receive from intermittent use, such as increased independence and the simulation of a regular voiding and filling cycle. We have revised the paragraph to reflect the reasoning behind this recommendation and added a reference comparing intermittent and indwelling catheters.

I miss CA-ASB in the part of CA-UTI 

Response: Thank you for this comment. We have revised the paragraph to indicate that CA-ASB is also common in catheterized patients.

- For the manuscript table 4 should be about CA-UTI. However, what is uncomplicated cystitis in patients with a catheter? Could the author clarify this part, and also include CA-UTI in the title of table 4 if this is about uropathogens in CA-UTI.

Response: This table is meant to reflect the distribution of pathogens in uncomplicated and complicated UTI. We have revised the associated paragraph to clarify that the pathogens seen in CA-UTI are largely the same as those seen in complicated UTI.

- In the pathogenesis part I miss a part about lactobacilli and the urinary microbiome.

Response: Thank you for this note. Lactobacilli are mentioned in the introduction, but we have added an additional reference to highlight the importance of these bacteria to the urinary microbiome. This review covers changes in abundance of Lactobacilli in a variety of bladder conditions.

Chorbińska, J.; Krajewski, W.; Nowak, Ł.; Małkiewicz, B.; Del Giudice, F.; Szydełko, T. Urinary Microbiome in Bladder Diseases—Review. Biomedicines 2023, 11.

- Where is figure 1?

Response: Thank you for pointing this error out. We have changed the number on Figure 2 to be Figure 1, since this is the only figure.

- Page 5 referral to Table 1 is not correct.

Response: We have adjusted this to reflect the correct table.

- For clearance, it would be optional to included subtitles, such as treatment / pathogenesis

Response: Thank you for this note. We have not added subheadings, but we have revised some of the topic/concluding sentences to add additional clarity.

Reviewer 2 Report

Comments and Suggestions for Authors

Dear authors

happy day

The paper is informative and contain valuable information. Meanwhile, it need some work.

Kindly do the following suggestions to improve the paper.

1- Kindly did not give your opinion through the text as a solid information.

Kindly either add references or use words that mean that you conclude that, such as apparently, it can be concluded that etc.

2- The introduction part need some work to rearrange the information and to be like a revision for the old and existed arts.

3- The aim of the review is mostly described once in the abstract and one time in the end of the introduction. However, if a new part is need a sort of highlighting, kindly be sure that you did not describe that before.

4- I feel that you jump from point to point. Kindly be sure that each paragraph contain one point and prepare the reader to the next idea in the end of the paragraph.

5- You need to assist your review with more references in each part where there is an information and the reference is missed.

6- Be so specific while you write.

In general the paper need to be re-written in more precise way, references need to be added in the correct place and opinion should be avoided or supported with references.

With my pleasure

Author Response

Thank you for providing these helpful comments.

1- Kindly did not give your opinion through the text as a solid information.

Kindly either add references or use words that mean that you conclude that, such as apparently, it can be concluded that etc.

Response: Thank you for this comment. We have reviewed the text and adjusted the language at points, as highlighted in the revised document.

2- The introduction part need some work to rearrange the information and to be like a revision for the old and existed arts.

Response: We have added a sentence to ensure the point of the review is clear to the reader.

  • Currently available literature lacks comprehensive reviews on the most common infections and inflammatory conditions affecting the lower urinary tract.

3- The aim of the review is mostly described once in the abstract and one time in the end of the introduction. However, if a new part is need a sort of highlighting, kindly be sure that you did not describe that before.

Response: Thank you for the comment. We have reviewed the writing for instances of repetition.

4- I feel that you jump from point to point. Kindly be sure that each paragraph contain one point and prepare the reader to the next idea in the end of the paragraph.

Response: Thank you for these comments. We have reviewed the manuscript and made some edits to reduce repetition and ensure clarity. We have also revised the topic and concluding sentences in the paragraphs to clearly bridge between concepts.

5- You need to assist your review with more references in each part where there is an information and the reference is missed.

Response: We have reviewed the manuscript and added additional references where they may be useful to support the information.

6- Be so specific while you write.

In general the paper need to be re-written in more precise way, references need to be added in the correct place and opinion should be avoided or supported with references.

Response: We have reviewed the manuscript for areas of improvement, ensuring all necessary references are present.

Reviewer 3 Report

Comments and Suggestions for Authors

Few comments:

This article requires a « List of Abbreviations and Acronyms”

To be explain: page 13 « inappropriate … agricultural use”

Where is the figure 1?

Who is XX (supervision)? (Author contribution page16)

Author Response

Thank you for providing these helpful comments.

This article requires a « List of Abbreviations and Acronyms”

Response: We agree with the reviewer and have added a list of abbreviations at the end of the manuscript.

To be explain: page 13 « inappropriate … agricultural use”

Response: We have revised this sentence to no longer mention agricultural use, since it is outside of the scope of this review.

Where is the figure 1?

Response: Thank you for pointing out this error. We have revised the figure numbers.

Who is XX (supervision)? (Author contribution page16)

Response: We have revised this section to indicate the correct division of labor. XX has been revised to C. L..